# A Cross-Sectional Study on Canine and Feline Anal Sac Disease

**DOI:** 10.3390/ani12010095

**Published:** 2021-12-31

**Authors:** Ronald Jan Corbee, Hilde H. Woldring, Lianne M. van den Eijnde, Erik G. H. Wouters

**Affiliations:** 1Department of Clinical Sciences, Faculty of Veterinary Medicine, Utrecht University, Yalelaan 108, 3584 CM Utrecht, The Netherlands; hildewoldring@hotmail.com (H.H.W.); liannevandeneijnde@hotmail.com (L.M.v.d.E.); 2AniCura Dierenziekenhuis Drechtstreek, Jan Valsterweg 26, 3315 LG Dordrecht, The Netherlands; erik.wouters@anicura.nl

**Keywords:** impaction, sacculectomy, dog, cat, gland, allergy, skin, diarrhea, non-neoplastic, inflammation

## Abstract

**Simple Summary:**

Anal sac disease is a common problem in private practice, but there is surprisingly little information available about anal sac disease in the literature. In this article, the incidence, predisposing factors, diagnostics, treatment options, and recurrence rates were investigated by the use of a questionnaire which was distributed among veterinarians in private practice. Dogs were more commonly affected than cats. Diarrhea and skin problems increased the risk for anal sac disease, and certain breeds were more often affected. Diagnosis was made based on the presence of clinical signs and characteristics of the anal sac content. Manual expression and treating any potential underlying disease were the most important treatments. Surgical removal was performed in the case of frequent recurrence. Surgical outcome of anal sacculectomy can be improved when surgery is performed after medical management. Future studies should investigate if these findings reported by veterinarians can be confirmed by asking veterinarians to keep a logbook on dogs and cats with anal sac disease.

**Abstract:**

Limited data are available on canine and feline non-neoplastic anal sac disease. Therefore, the aim of this study was to obtain observational data on the incidence, predisposing factors, diagnosis, treatment, and recurrence rate of canine and feline anal sac disease. To this end, a questionnaire was distributed among veterinarians. The incidence of non-neoplastic anal sac disease was estimated at 15.7% in dogs and 0.4% in cats. Predisposing factors were diarrhea, skin problems, several dog breeds, and particularly small breed dogs, male cats, British shorthairs, and obesity in dogs. Diagnosis was made based on the presence of clinical signs and characteristics of the anal sac content. Manual expression and treating any potential underlying disease were the most important treatments for all three types of non-neoplastic anal sac disease. Anal sacculectomy was performed in refractory cases. The most recurrent anal sac disease condition was impaction. Diagnosis of anal sac disease should be based on clinical signs and rectal examination, as the evaluation of the anal sac content is not reliable. Surgical outcomes of anal sacculectomy can be improved when surgery is performed after medical management. Future studies should investigate these findings in prospective trials.

## 1. Introduction

Anal sac disease can be divided into neoplastic and non-neoplastic conditions (also called inflammatory anal sac disease) [1,2]. The present study focusses on non-neoplastic conditions, which can be divided into three types: impaction, inflammation, and abscessation [3,4]. An anal sac impaction is characterized as an enlargement of the sacs due to retention of anal sac content, without showing any signs of inflammation except for pain and discomfort [3]. The sacs are filled with thick content that can be difficult to express [5]. Anal sac impaction can occur unilaterally, but most often it presents itself as bilateral [3]. Anal sac inflammation, also called anal sacculitis, is defined as an enlargement combined with inflammation of the anal sac [3]. The anal sac and perianal region often become red, swollen, and painful [5]. In the case of an anal sac abscess, pyrexia is often present; however, pyrexia can also occur in the case of severe anal sac inflammation. Furthermore, an inflamed and often alopecic area of the anal sacs can be observed in the case of an anal sac abscess, again accompanied with swelling and pain. Additionally, discharge contaminated with blood or purulent material can occur [1,5]. As an anal sac abscess progresses, the abscess can rupture and draining fistulas can develop that may be noted on the perineum [3,5]. A few studies determined the incidence of anal sac disease in dogs, which varied between 4.9% and 12.5% in different subpopulations [6,7,8].

The incidence of anal sac disease in cats has not been reported to the authors’ knowledge, but it is stated that anal sac disease is more common in dogs compared to cats [2,9]. Several breeds have been indicated as being predisposed to anal sac disease, such as Labrador Retrievers, German Shepherds, as well as several small breed dogs [2,7,8,10,11,12,13]. Other reported risk factors are the presence of diarrhea [3,14,15] as well as dermatological conditions [15,16].

Clinical symptoms of canine anal sac disease have been investigated before and include: “scooting”, discomfort when sitting down, licking/biting of the anal area or tail-base region, tail chasing, tenesmus, perianal discharge, redness of the tail area, moist dermatitis of the perianal region, and back rubbing against an object [2,4,10,17]. As other diseases such as parasites, flea allergies, perianal tumors, perianal fistulae, and pyoderma can cause similar symptoms [3,15], executing a rectal examination and careful palpation of the anal sacs are very important in making a diagnosis [3,5].

The clinical signs in cats suffering from anal sac disease have, to the authors’ knowledge, not yet been described in the literature, but were suggested to be similar. In the case of anal sac disease, a visible or palpable perianal swelling at four or eight o’clock lateral to the anus can be noted.

The distinction between anal sac impaction, inflammation, and abscessation can be vague and the criteria on how to distinguish between these conditions differ between studies. One study reported that the distinction must rely on history, clinical examination, cytology, and a bacterial culture [12], although another study stated that the distinction can be made by digital palpation of the anal sacs and by examining the anal sac content [13].

No randomized, controlled studies have been conducted yet on the most appropriate and effective treatment for dogs with anal sac impaction, anal sac inflammation, or anal sac abscessation. Suggested treatment options for anal sac impaction and inflammation are a combination of expressing the anal sacs, flushing the anal sacs, warm packing, high-fiber diet, administering anti-inflammatory drugs, infusing topical antibiotics or antibiotic-corticosteroid preparations, or prescribing systemic antibiotics [3,5,14,15]. In addition, treatment of possible underlying factors, such as dermatological and gastrointestinal diseases, is recommended [5,13]. In case an abscess has developed and the therapy is unsuccessful, an incision over the anal sac can be made, allowing for open drainage [5].

If all of the treatments are unsuccessful and the dog has recurrent anal sac disease or in severe cases in which a fistulous tract in an abscess persists, surgical resection of the anal sacs (anal sacculectomy) has to be considered. Several methods for anal sacculectomy have been described, including an open method, a closed method, a modified closed method, and a modified open method [11,18].

The prognosis for anal sac disease depends on the underlying diseases and the severity of clinical signs. However, overall, the prognosis is variable [13].

To provide more insight into several aspects of anal sac disease and how veterinary practitioners currently deal with this disease, the aim of this study was to obtain observational data on the incidence, predisposing factors, diagnosis, treatment, and recurrence rate of canine and feline anal sac disease.

## 2. Materials and Methods

### 2.1. Questionnaire

This observational study was conducted in the form of two online open surveys (one for each species), which were created in Qualtrics according to the Cherries Guidelines [19]. This study did not require evaluation by an ethical committee under Dutch legislation, as all veterinarians that participated did so on a voluntary basis and they could withdraw from this study at any time. We aimed at 50 participants per survey to obtain a sufficient number of observations [20]. The surveys consisted of 45 multiple choice questions and descriptive questions divided into six sections. Each page contained a maximum of six questions and the total number of pages was ten. The surveys are available in Appendix A.

No randomization of items or adaptive questioning to reduce number or complexity of the questions has been performed. Before distributing the surveys, the usability, quality, and completeness of the questions were tested during phone calls with three independent Dutch veterinarians. The technical functionality was tested by the investigators by previewing the questionnaire several times. While participating in the questionnaire, there were no mandatory items and no completeness check was performed before or after submitting the questionnaire. However, in the introduction prior to the survey, it was highly emphasized that for the quality of the study, it was necessary to answer all the questions completely. Most questions provided a non-response option, such as “other, please specify…”, but this was not the case for all the questions. If possible, selection of only one option was enforced. Participants were able to review and change their answers during the survey through a back button.

Since the survey was distributed by copying and pasting an anonymous link on Facebook (closed group for Dutch veterinarians) and in e-mail messages, the responses did not provide identifying information, such as an IP address or cookies, and multiple responses from the same person (ballot box stuffing) were not prevented. Therefore, the identification of unique visitors could not be determined. However, participants could leave their e-mail address if they wished to receive the final report and this e-mail address, together with the survey results, allowed for the identification of some unique participants. If there were multiple responses from the same participant, only the most recent response was kept for analysis. Some participants did not go through all of the questionnaire pages, because participants could discontinue the survey at any time, leaving one or more pages incomplete. However, both complete and incomplete questionnaires were collected and therefore, each given answer was analyzed. As a result, the number of answers per question varied.

At the first page of the questionnaire, there was informed consent. The participants were informed about the study design, which was an observational study. Moreover, the investigators were mentioned, as well as the institution from which this study was carried out. In addition, the purpose of the study was explained. The length of time for completing the survey was also stated and this was estimated to be ten to fifteen minutes. It was emphasized that personal data would be handled with great care and that the data would be deleted after processing. This personal information was collected and stored in the online survey, which was protected from unauthorized access by a password and two-factor authentication. Only the investigators had access to this.

The first section consisted of questions regarding the incidence of anal sac disease (subdivided into impaction, inflammation, and abscessation). In the second section, participants were asked about predisposing factors for dogs or cats with anal sac disease, including the gender, neuter status, age, body condition score, size, breed, coat, diet, presence of skin disorders, and presence of gastrointestinal disorders. There were also questions concerning whether there was an association between the type of season and the occurrence of clinical signs and whether there was an association between the presence of diarrhea and the type of diarrhea (small- or large-bowel diarrhea) and the occurrence of clinical signs.

In the third section, questions were asked about the diagnostics. First, the criteria used for the diagnosis of any kind of anal sac disease were asked for. Then, questions were asked on how the distinction between impaction, inflammation, or abscessation was made. In the fourth section, participants were asked about the treatment of anal sac disease. First, the treatment for impaction, inflammation, or abscessation in dogs was assessed.

Next, questions were asked about whether there is a difference in the treatment of impaction, inflammation, or abscessation of the anal sacs in cats, how the decision for anal sacculectomy is made, which method (open versus closed) is preferred, and how the procedure is carried out. In the fifth section, questions were asked regarding the effect of treatment, prognosis, and recurrence. In the final section, participants could give any comments or recommendations for the survey.

### 2.2. Data Analysis

All responses were collected from Qualtrics^®^ into Excel. The total number of responses was recorded thereafter and empty questionnaires, in which no single answer was given, were removed. In the end, the total number of complete and partially complete responses was recorded. Subsequently, the questions were separated and analyzed, whereby the number of answers for each separate question was recorded. Because both complete and incomplete questionnaires were collected, this number varied per question.

The mean incidence of anal sac disease in the dogs and cats and the mean incidence for each individual anal sac condition (impaction, inflammation, and abscessation) were calculated separately based on the number of dogs or cats with anal sac disease and the total number of dogs or cats seen by the participant. First, the answers were checked to ascertain if they were all complete and correct. If responses were empty or if the data were considered incorrect, they were removed from the dataset. When a range was given, the average was included in the dataset.

Of the remaining responses, the number of dogs or cats with anal sac impaction was added to the number of dogs or cats with anal sac inflammation and abscessation to check if this number was equal to the total number of dogs or cats with anal sac disease. If this number differed, a new total number of dogs or cats with anal sac disease was calculated by adding the number of dogs or cats with anal sac impaction to the number of dogs or cats with anal sac inflammation and anal sac abscessation.

For the average time for recurrence of anal sac disease and the average time for recurrence of anal sac impaction, anal sac inflammation, and anal sac abscessation, the answers were also checked to ascertain if they were complete and correct. If responses were empty or if the data were considered incorrect, they were removed from the dataset. When a range was given, the average was included in the dataset. If a range included “a few months” or “a few weeks”, this was calculated as three months or three weeks.

As the generated data were only descriptive, no statistical analysis has been performed.

## 3. Results

### 3.1. Number of Surveys

For the canine survey, 69 responses were received, of which 50 (72.5%) were complete, and 19 (27.5%) were partial responses. The feline survey had 78 responses, of which 48 (61.5%) were complete, and 30 (38.5%) were partial responses.

### 3.2. Incidence

The incidence of anal sac disease in dogs and cats is given in Table 1.

### 3.3. Predisposing Factors

In dogs and cats, gender, neuter status, type of coat, and diet were not regarded as predisposing factors. Age was a factor, as anal sac disease was reported to occur more often in adult dogs and cats compared to puppies and kittens (most of the participants reported occurrence in adults as more common (87.7%, *n* = 50 in dogs, 67.8%, *n* = 40 in cats); the others mentioned no difference).

Almost one-third of the participants (31.6%, *n* = 18 in dogs, 36.1%, *n* = 22 in cats) reported that anal sac disease is more prevalent in obese animals, while a few participants (1.8%, *n* = 1 in dogs, 9.8%, *n* = 6 in cats) reported that anal sac disease is more prevalent in non-obese animals. However, the majority of the participants (66.7%, *n* = 38 in dogs, 54.1%, *n* = 33 in cats) reported no difference.

Breed size distribution is given in Table 2.

Dog breeds in which anal sac disease was reported the most are given in Table 3.

In cats, participants noted a higher occurrence of anal sac disease in domestic shorthairs (37.7%, *n* = 23) and British shorthairs (16.4%, *n* = 10).

Anal sac disease in dogs seems to be more prevalent in spring/summer as was mentioned by almost a quarter of the participants (25.5%, *n* = 13), while almost three-quarters of participants (74.5%, *n* = 38) reported that there is no difference in occurrence between spring/summer and autumn/winter. No participants reported that anal sac disease in dogs is more common in autumn/winter. In cats, no differences in occurrence by season were reported.

Cutaneous adverse food reactions or food allergies (37.9%, *n* = 39 in dogs, 20.7%, *n* = 12 in cats) and atopic dermatitis (30.1%, *n* = 31 in dogs, 10.3%, *n* = 6 in cats) were dermatological conditions mentioned as being associated with anal sac disease.

Adverse food reactions (33.7%, *n* = 29 in dogs, 17.9%, *n* = 10 in cats) and viral or bacterial enteritis (25.6%, *n* = 22 in dogs, 10.7%, *n* = 6 in cats) were mentioned as gastrointestinal diseases associated with anal sac disease. About half of the participants (48.1%, *n* = 26) reported that there is a relation between the presence of diarrhea and the occurrence of anal sac disease in dogs, but that there is no difference between small- and large-bowel diarrhea. Almost a quarter of the participants (25.9%, *n* = 14) reported that there is no difference between the presence of diarrhea and the occurrence of anal sac disease in dogs. A minority of participants reported that there is a relation between the occurrence of small- or large-bowel diarrhea and the presence of anal sac disease in dogs; 11.1% (*n* = 6) and 14.8% (*n* = 8) reported that anal sac disease is more prevalent in dogs with small- and large-bowel diarrhea, respectively.

In cats, the majority of participants (70.7%, *n* = 41) reported no relation between the occurrence of anal sac disease and diarrhea.

### 3.4. Diagnosis

Clinical signs were important for diagnosing anal sac disease (87.0%, *n* = 47 in dogs, 80.0%, *n* = 40 in cats), followed by the size of the anal sac (72.2%, *n* = 39 in dogs, 80.0%, *n* = 40 in cats), the nature of the anal sac content (81.5%, *n* = 44 in dogs, 68.0%, *n* = 34 in cats), the consistency of the anal sac content (74.1%, *n* = 40 in dogs, 66.0%, *n* = 33 in cats), the presence of pain at palpation (77.8%, *n* = 42 in dogs, 60.0%, *n* = 30 in cats), the amount of anal sac content (64.8%, *n* = 35 in dogs, 64.0%, *n* = 32 in cats), the color of the anal sac content (66.7%, *n* = 36 in dogs, 54.0%, *n* = 27 in cats), the consistency of the anal sac (68.5%, *n* = 37 in dogs, 50.0%, *n* = 25 in cats), the ease of emptying the anal sac (61.1%, *n* = 33 in dogs, 46.0%, *n* = 23 in cats), the color of the anal area (40.7%, *n* = 22 in dogs, 22.0%, *n* = 11 in cats), the smell of the anal sac content (33.3%, *n* = 18 in dogs, 24%, *n* = 12 in cats), the body temperature (25.9%, *n* = 14 in dogs, 22.0%, *n* = 11 in cats), the shape of the anal sac (24.1%, *n* = 13 in dogs, 22.0%, *n* = 11 in cats), the reaction on inserting the thermometer (13.0%, *n* = 7 in dogs, 8.0%, *n* = 4 in cats), the temperature of the anal sac (9.3%, *n* = 5 in dogs, 2.0%, *n* = 1 in cats), and microscopic examination of the anal sac content (1.9%, *n* = 1 in dogs, 6.0%, *n* = 3 in cats). Further distinction between impaction, inflammation, and abscessation is based on the anal sac content, discomfort, and fistulation by most participants for both dogs and cats (Appendix A).

### 3.5. Treatment

All participants treated anal sac impaction by emptying the anal sac by manual expression (100%, *n* = 53) in dogs, and almost all participants (94%, *n* = 47) used this treatment for anal sac impaction in cats. Treating an underlying cause was the second most common treatment for anal sac impaction (47.2%, *n* = 25 in dogs, 34.0%, *n* = 17 in cats).

For anal sac inflammation, emptying the anal sac by manual expression was still the main treatment (88.7%, *n* = 46 in dogs, 89.8%, *n* = 44 in cats), followed by treating an underlying cause (50.9%, *n* = 27 in dogs, 40.8%, *n* = 20 in cats), administration of antibiotic ointment in the anal sac (34.0%, *n* = 18 in dogs, 20.4%, *n* = 10 in cats), systemic antibiotics (34.0%, *n* = 18 in dogs, 42.9%, *n* = 21 in cats), flushing the anal sacs (30.2%, *n* = 16 in dogs, 30.6%, *n* = 15 in cats), prescription of non-steroidal anti-inflammatory drugs (28.3%, *n* = 15, 12.2%, *n* = 6 in cats), and prescription of corticosteroids (18.9%, *n* = 10 in dogs, 0.0% *n* = 0 in cats).

Abscessation was treated by prescribing a systemic antibiotic (90.6%, *n* = 48 in dogs, 72.0%, *n* = 36 in cats), manual expression of the anal sacs (56.6%, *n* = 30 in dogs, 70.0%, *n* = 35 in cats), and flushing the anal sacs (43.4%, *n* = 23 in dogs, 46.0%, *n* = 23 in cats). Non-steroidal anti-inflammatory drugs (28.3%, *n* = 15 in dogs, 14.0%, *n* = 7 in cats) and drainage (11.3%, *n* = 6 in dogs, 14.0%, *n* = 7 in cats) were less frequently mentioned for treatment of anal sac abscessation. More information on type of flush, type of antibiotic ointment, and type of systemic antibiotic prescribed can be found in Appendix A.

### 3.6. Anal Sacculectomy

Frequent recurrence was mentioned by veterinarians as the most chosen reason to proceed to surgical removal of the anal sacs (75.0%, *n* = 36 in dogs, 69.6%, *n* = 16 in cats). Two different methods were used for surgical removal of anal sacs, the open and closed method. The closed method was most often used (76.7%, *n* = 23 in dogs, 61.1%, *n* = 13 in cats). Surgery was not performed by 38.3% of the participants (*n* = 18) in dogs, and by 61.7% of the participants (*n* = 47) in cats.

### 3.7. Recurrence Rate

The mean percentages of dogs with a relapse of anal sac impaction, inflammation, or abscessation were 35.7%, 6.3%, and 2.9%, respectively. In cats, these percentages were higher, at 40.5%, 30.1%, and 17.8%, respectively. In dogs, recurrence of anal sac impaction and inflammation occurs on average after 4–5 months; for abscessation, the mean recurrence time is reported to be 10 months on average. In cats, recurrence usually occurs on average after 4–5 months for all forms of anal sac disease.

## 4. Discussion

The incidence of anal sac disease was higher in dogs (15.7%), compared to cats (0.4%), which is in agreement with statements in previous studies [2,9]. This can be explained by the difference in composition of anal sac content and location of the anal duct opening. In dogs, the apocrine glands are more abundant compared to the sebaceous glands. Whereas in cats, these glands are more equally distributed, in location as well as in number. Sebaceous glands produce a more lipid secretion, which results in more lipid anal sac content. Combined with the more lateral location of the anal duct opening in cats, this could explain the lower incidence of anal sac disease in cats [2,14,15].

To the authors’ knowledge, this is the first study to report incidence numbers of anal sac disease in cats. Whether the incidence numbers in this study, based on self-reported retrospective data, are correct needs to be proven in a prospective trial in which case logs are kept. In the present study, the most frequent anal sac disease was anal sac impaction (8.9% in dogs, 0.2% in cats) followed by inflammation (4.1% in dogs, 0.1% in cats) and abscessation (2.8% in dogs, 0.1% in cats). An incidence of 7.1% for anal sac impaction has been reported in a previous study in 3884 dogs from 89 clinics, where anal sac disease was regarded as one of the most frequently recorded disorders in primary veterinary practices in England [7]). In addition, the incidence of 8.9% is also in accordance with a study in which 8.8% (*n* = 49) of 559 dogs with dermatological conditions were diagnosed with anal sac impaction [21]. However, the incidence of anal sac impaction is expected to be higher in dogs with dermatological conditions, suggesting a higher overall incidence in our study.

In the present study, anal sac disease was reported to be more prevalent in adult or old dogs (>1 year). This is similar to a previous study where the mean age of dogs with anal sac disease was reported to be 5.6 years (±2.8 years), which corresponds with the response category “adult to old dogs” in the present study, suggesting that adult dogs as well as older dogs may be at increased risk, while anal sac disease in young dogs (<1 year) is less common [10].

Based on the present study, the same can be concluded for cats, which is in agreement with the previous findings that cats younger than one year appeared to have significantly more watery anal sac secretions in contrast to older cats. This may contribute to the higher occurrence of anal sac disease observed in mature cats, since watery secretions are easier to dispose of [4].

Obesity was reported as a risk factor in both species. This is in accordance with other studies stating that an increased body condition score is a risk factor for anal sacculitis [22] or anal sac disease [14,15]. Perianal fat can lead to compression of the anal sac ducts, resulting in retention of anal sac content [14,15]. Furthermore, obese dogs may have poor anal sphincter muscle tone [3]. Anal sac disease was reported to be more common in small dogs (<10 kg) and medium-sized dogs (between 10 and 30 kg). These findings are in accordance with a previous study reporting the weight of dogs with anal sac disease to be 14.0 ±10.5 kg [10]. This could be due to relatively small anal sac ducts that can easily become obstructed if the anal sac content becomes more thickened [15], or because of poor anal sphincter muscle tone in small breed dogs [3].

The reported breeds from the present study are comparable to breeds observed in previous studies, most of which are small breed dogs [7,10]. As reported previously, the Chihuahua might be even more at risk of developing anal sac disease when compared to other small breeds [8]. Although the Labrador Retriever is a large breed, this breed also seems to be at higher risk of developing anal sac disease which was also observed in a previous study showing that Cavalier King Charles Spaniels and Labradors or Labrador crossbreeds were overrepresented in the dog population presented for anal sacculectomy [11].

Labradors are more often overweight or obese [23], which is a predisposing factor as demonstrated earlier. In addition, the Labrador Retriever is known to be at increased risk for developing a food allergy or sensitivity [24], which might be a risk for developing anal sac disease as well, as recurrent anal sac disease has been found in dogs with dermatological conditions including food hypersensitivity [3,13].

The German Shepherd dog might well be at increased risk for developing anal sac disease, since the anal sacs of this breed, compared to other breeds, lie deeper in the perianal tissues near the rectum, which could be a risk for impaction or infection [15]. In the present study, 5.2% of the participants reported that anal sac disease occurs more often in the German Shepherd.

As the Labrador Retriever, German Shepherd, and Cavalier King Charles Spaniel all belong to the top 10 most popular dogs in the Netherlands, it is unclear whether this is a true predisposition, just as in cats, where the domestic shorthair, which was mentioned as the breed with the highest incidence of anal sac disease, composes by far the largest number of feline patients in primary practice. British shorthair cats were also mentioned, which are known for their higher incidence of obesity [25].

Season has a small influence on the incidence of anal sac disease in dogs. The increased incidence of anal sac disease during the spring/summer might be explained by the relationship between anal sac disease and canine atopic dermatitis which was also demonstrated in the present study [3,13].

The anal sacs can be considered as part of the skin [1]. If the skin is affected as a result of a cutaneous food reaction or atopic dermatitis, the skin within the anal sacs is also affected, resulting in increased secretion of fluid, glandular hypersecretion, and subsequent occlusion of the ductal opening and infection. In cats, the relationship with season was not demonstrated, probably because coinciding cutaneous food reaction and atopic dermatitis were less reported in cats.

Adverse food reaction, as well as bacterial enteritis, can result in diarrhea or soft stool, preventing the natural expression of the anal sacs during defecation which in turn causes prolonged retention and occlusion of the anal sacs [17]. Moreover, diarrhea can cause irritation and swelling of the perineal region, which can also lead to occlusion of the opening of the anal sac duct and impaction, inflammation, or abscessation subsequently. However, not all dogs with soft feces or diarrhea develop anal sac disease [10].

In cats in the present study, dermatological and gastrointestinal diseases were less frequently reported to be associated with anal sac disease.

According to previous studies, diagnosis of anal sac disease should be based on the presence of clinical signs and the findings from a clinical and rectal examination [1,3]. In the present study, diagnosis was primarily based on the presence of clinical signs in both dogs and cats. Other diagnostics that were used, i.e., the nature of the anal sac content, the consistency of the anal sac content, the amount of anal sac content, the color of the anal sac content, the smell of the anal sac content, and microscopic examination of the anal sac content, are not very reliable.

Normal anal sac secretions can be liquid or pasty, with a variation in normal color. There can also be large numbers of leukocytes in the normal anal sac secretion. Moreover, the type and amount of yeasts and bacteria seen in healthy dogs and cats and dogs and cats with anal sac disease are similar [3,4,9,10,16,26]. In addition, a large population of leukocytes or the presence of numerous bacteria in the anal sac content can also be a normal finding. As a consequence, macroscopic characteristics, cytology, and bacteriology cannot be used as diagnostic tools to distinguish between healthy dogs and dogs with anal sac disease [10], nor to distinguish between anal sac impaction, inflammation, and abscessation [1].

Anal sac inflammation is characterized by a swollen, red, and painful perianal area. Additionally, in the case of anal sac inflammation, the anal sac content can contain blood [5]. In the case of an anal sac abscess, the sac is enlarged and the skin overlying the anal sacs will become thin, erythematous, and edematous [1,15]. The anal sac content can be observed as bloody/purulent in the case of an anal sac abscess, but also in the case of severe anal sac inflammation [5]. Furthermore, the sacs are extremely painful in the case of an anal abscess [5]. Pyrexia and fistulation are also indicative of an anal sac abscess.

To the authors’ knowledge, there are no clinical trials that have evaluated the effects of treatment options for anal sac disease. Treatment options mentioned are based on clinical experience, and are mostly in agreement with suggestions from previous studies. Anal sac impaction can often be treated solely by manual expression of the anal sac content [1,3,13]. In addition, treating the underlying cause such as a skin disease or diarrhea is an important part of the treatment [5].

Manual expression of the anal sac content is also advised in the case of anal sac inflammation [3,5,13]. However, because inflammation can be more painful, sedation or general anesthesia may be necessary. In addition, other treatments that are described in studies may be necessary as well, depending on the severity of the condition. These treatments include flushing the anal sacs using an antiseptic solution, warm packing of the anal sacs, administering anti-inflammatory drugs, or applying a local antibiotic or antibiotic-corticosteroid ointment [1,5,13], which is in accordance with the responses from the participating veterinarians in the present study. Only in severe cases, for example when pyrexia or other systemic signs are present, are systemic antibiotics indicated [3,13].

In the case of anal sac abscessation, systemic antibiotics are recommended, as well as emptying and flushing the anal sacs and applying antibiotic ointment [13] and drainage [5]. Flushing is performed using a saline solution [3] or an antiseptic solution such as a combination of chlorhexidine or povidone-iodine solution added to saline [1,5]. Based on literature, chloramphenicol should be the first choice as antibiotic ointment in anal sac disease [27,28]. Sulfadimidine, which is a sulfonamide just like sulfadiazine, is effective in vivo although in vitro the bacteria seem to be resistant. Therefore, silver sulfadiazine may not be an appropriate choice [27]. In the case of systemic antibiotics, spiramycin and metronidazole should be the first choice, followed by amoxicillin [28]. In cases of pyrexia and/or pain, non-steroidal anti-inflammatory drugs or corticosteroids can be prescribed.

To prevent healthy dogs from experiencing anal sac impaction or to prevent a relapse in previously impacted anal sacs, sometimes anal sacs are routinely expressed by a groomer or veterinarian [10,13]. This should be performed carefully as it can also initiate irritation of the anal sacs and surrounding tissues, which might even enhance recurrences or progression to infection [3,13]. A high-fiber diet is also recommended for prevention of anal sac disease [3,14,15].

Surgical removal of anal sacs is indicated in the case of frequent relapse of anal sac disease despite appropriate treatment [5,11,15], in case there is no response to therapy, in case a patient has a fistulous tract that persists, or in case a patient has anal sac abscessation that does not respond to open drainage [1,3,5,29]. The surgical method that was preferred by most of the participants is the closed method. This was also recommended in previous studies, which stated that an open method is significantly associated with a higher number of complications, including contamination of the surrounding tissues and fecal incontinence [30,31]. It is recommended to proceed to surgical removal after the third pharmacological treatment of anal sac impaction, inflammation, or abscessation. Current inflammation or abscessation, irrespective of time of onset of the disease, makes anal sacculectomy more complicated.

Although recurrence rates were higher in cats compared to dogs, anal sacculectomy was performed more frequently in dogs because veterinarians indicated that they were less experienced in the surgical procedure in cats. Anal sac impaction has the highest recurrence rate, with a reported time of recurrence of 4–5 months. This is in contrast with a previous study among pet owners who reported that clinical signs recurred within a median frequency of three weeks (ranging from one to three days to one to two months) or a mean of 2.5 weeks after anal sac expression. The majority of the dogs (81.3%) only stopped scooting for three weeks or less, while 18.8% of the dogs stopped scooting for only one to three days [10].

The present study demonstrated several aspects of anal sac disease in dogs and cats, and how it is dealt with in private practice. All the data were self-reported by veterinarians in a retrospective view. A prospective trial should give more insight into the true incidence, risk factors, and recurrence rates of anal sac disease in dogs and cats. Furthermore, clinical trials should be performed to evaluate the effect of different treatment options in order to ultimately provide evidence-based recommendations for treatment and prevention of this common disorder in dogs (and cats).

## 5. Conclusions

Anal sac disease is a common disorder in dogs, and it also occurs in cats. Diagnosis should be based on clinical signs and rectal examination, and not on the nature or consistency of the anal sac content. It is important to distinguish impaction, inflammation, and abscessation, because they require different treatment. Anal sacculectomy should be performed after medical treatment to improve the surgical outcome. Prospective clinical trials are needed to be able to provide evidence-based recommendations for treatment and prevention of anal sac disease in dogs and cats.

## Figures and Tables

**Table 1 animals-12-00095-t001:** Incidence of anal sac disease and the subdivision between anal sac impaction, anal sac inflammation, and anal sac abscess in dogs and cats (based on 56 and 57 responses, respectively).

Condition	Incidence in Dogs	Incidence in Cats
Anal sac disease	15.70%	0.38%
Anal sac impaction	8.90%	0.25%
Anal sac inflammation	4.05%	0.06%
Anal sac abscess	2.75%	0.07%

**Table 2 animals-12-00095-t002:** Reported predisposition to anal sac disease per breed size in dogs.

Breed Size	Percentage	Number of Responses
Small (<10 kg)	60.34%	35
Medium (10–30 kg)	17.24%	10
Large (>30 kg)	3.44%	2
No difference	19%	11

**Table 3 animals-12-00095-t003:** Reported predisposition of anal sac disease per dog breed.

Breed	Percentage	Number of Responses
Chihuahua	14.53%	25
Labrador Retriever	12.79%	22
French Bulldog	11.63%	20
Jack Russell Terrier	10.46%	18
Lhasa Apso	8.14%	14
Beagle	6.40%	11
German Shepherd	5.23%	9
Boomer	4.07%	7
Cocker Spaniel	3.49%	6
Miniature Poodle	2.91%	5
Staffordshire Bull Terrier	2.91%	5
Shih Tzu	2.91%	5
Cavalier King Charles Spaniel	2.91%	5
West Highland White Terrier	1.74%	3
Springer Spaniel	1.74%	3
Maltese	1.16%	2
Labradoodle	1.16%	2
German Shorthaired Pointer	0.58%	1
No difference	5.23%	9

## Data Availability

All the generated data will be available upon reasonable request.

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
