# Peer review of "A Cross-Sectional Study on Canine and Feline Anal Sac Disease"

_animals, 2021, doi:10.3390/ani12010095_

Round 1

Reviewer 1 Report

This article was well written, However, looking at the overall structure of this article, several problems were found.

The thesis was full of long, rather wordy and narrative sentences.  It would be better if the sentences were more concise and academic.  In addition, the authors should write a maximum of 5 sentences per paragraph and make each paragraph distinct and visually clear.

I think that authors should organize their data analysis or subject inclusion procedures in flowcharts or tables so that readers can easily understand them. In addition, there are too many numbers in the sentences in the “Results” and “Discussion” sections.  It will be helpful to organize these sections using tables or graphs and subsection headings. 

Please list only what the authors discovered and learned using the questionnaire.   Also, the authors should briefly describe the statistical methods used to analyze the data.

Author Response

Thank you for your suggestions.

Sentences were shortened and paragraphs added to ease readability.

Suggestions on presenting data in tables and figures were taken into account based on the comments of reviewer 2.

The data presented and discussed were all generated from the questionnaire (as their was an open field to fill in other comments and suggestions). In the discussion we added the scientific backup/critical review, to provide the readers with guidelines on how to better diagnose, treat, and prevent canine and feline anal sac disease. We would therefore like to keep this information in.

"As the generated data were only descriptive, no statistical analysis has been performed." added to the text to clarify

Reviewer 2 Report

Diseases of the anal sac are a significant problem in veterinary medicine. The authors made the research with the use of an on-line questionnaire. The downside of this type of research is to start a conversation with the respondent in the event that he or she has a problem answering the question. It is revealed in the number of unfinished questionnaires. The results are interesting and relevant. They should be presented in tabular form, which would allow for easier interpretation of the results. The breed of the patient and the percentage of cases should be included in the table. Data on age and maintenance may remain in the current form.

paragraph 408- it's hard to give the anal sac ointment chloramfenicol- please explain.

paraghaph 412- there is no need to add a comment on clavulonic accid.

Some of the comments are included in the attached text.

Author Response

Thank you for your valuable comments.

We did a try-out of the questionnaire, and it seemed to be clear. Usually, participants drop off due to the length of the questionnaire, however if this was the case has not been recorded, so this remains speculative and is therefore not mentioned in the manuscript.

Breed and percentage of cases are now shown in tables, as suggested.

408: it is mentioned as first choice antibiotic, not that it is hard to give

412: removed as requested

comments in the attachemnet are now added to the text